# A simulation study of the use of temporal occupancy for identifying core and transient species

Sara Snell Taylor[1]*, Jessica R. Coyle[2], Ethan P. White[3,4], Allen H. Hurlbert[1,5]

1 Department of Biology, University of North Carolina, Chapel Hill, North Carolina, United States of America,
2 Department of Biology, Saint Mary's College of California, Moraga, California, United States of America,
3 Department of Wildlife Ecology and Conservation, University of Florida, Gainesville, Florida, United States of America, 4 Informatics Institute, University of Florida, Gainesville, Florida, United States of America,
5 Curriculum for the Environment and Ecology, University of North Carolina, Chapel Hill, North Carolina, United States of America

* ssnell6@gmail.com

**Data Availability Statement:** All scripts, outputs, and example data for this manuscripts are available on Github at https://github.com/ssnell6/CT-sim, which is linked to a parent repository https://github.com/hurlbertlab/core-transient-simulation

## Abstract

Transient species, which do not maintain self-sustaining populations in a system where they are observed, are ubiquitous in nature and their presence often impacts the interpretation of ecological patterns and processes. Identifying transient species from temporal occupancy, the proportion of time a species is observed at a given site over a time series, is subject to classification errors as a result of imperfect detection and source-sink dynamics. We use a simulation-based approach to assess how often errors in detection or classification occur in order to validate the use of temporal occupancy as a metric for inferring whether a species is a core or transient member of a community. We found that low detection increases error in the classification of core species, while high habitat heterogeneity and high detection increase error in classification of transient species. These findings confirm that temporal occupancy is a valid metric for inferring whether a species can maintain a self-sustaining population, but imperfect detection, low abundance, and highly heterogeneous landscapes may yield high misclassification rates.

## Introduction

Understanding the processes underlying community assembly is one of the primary goals of community ecology. Traditional approaches make inferences about community processes based on the set of species identified as community members, typically those observed at a study site [1, 2]. Data on communities are typically gathered via field surveys at a given site for one or more time points. However, the record of species from such community surveys often includes transient species that do not maintain self-sustaining populations in that community [3]. A growing number of studies use temporal occupancy, or the proportion of a multi-year time series over which a species is observed, to determine which species are "core" members of their communities and which species are transient [3–9]. Temporal occupancy provides a

containing all simulation functions to allow for replication of our simulation study.

**Funding:** This research was supported the National Science Foundation through grant DEB-1354563 to A. H. Hurlbert and E. P. White and by the Gordon and Betty Moore Foundation's Data-Driven Discovery Initiative through Grant GBMF4563 to E. P. White. The funders had no role in study design, data collection and analysis, decision to publish, or preparation of the manuscript.

**Competing interests:** The authors have declared that no competing interests exist.

quantitative measure of persistence within a community over time and its distribution tends to be bimodal [3]. Previous studies have used arbitrary thresholds (e.g. core species are those that occur in more than 50%, or 67%, or 75% of years [3, 5, 7], and Snell Taylor et al. (2018) found that a wide range of ecological patterns were generally robust to the specific threshold used. Nevertheless, ecological data collection is imperfect and using temporal occupancy to infer core or transient classification is susceptible to classification error.

One type of error is inferring that a species is transient when it is actually a core member of the community. A self-sustaining species that is present on the landscape every year may fail to be observed in some years, and hence be misclassified as a transient species, for three primary reasons (Table 1). These missed detections can occur due to low population densities [10–12], less conspicuous morphology (e.g., drab plumage) or behavior (e.g., singing quietly or infrequently; [13, 14]), and habitat structure with characteristics that limit the distance over which individuals can be detected (e.g., dense vegetation which may obscure sightings and attenuate sound; [15–18]). This latter possibility in particular may lead to potentially confounding gradients of average detectability along large-scale environmental gradients that range from open, low productivity deserts and grasslands to higher productivity forests. Although the effect of imperfect detectability on temporal occupancy and species classification is qualitatively understood, it is unclear how frequently and at what levels of detectability and abundance such errors occur.

The opposite classification error is also possible, where a species is inferred to be a core member of a community based on frequent occurrence in a time series, even though it does not maintain a locally viable population (Table 1). Some individuals of a species are observed regularly in habitats in which they do not successfully reproduce by dispersing in from adjacent suitable habitat [19, 20]. For example, in plants, seeds might be regularly dispersed into inhospitable habitats [21] and in birds, younger and lower quality males are often displaced by dominant males to adjacent, suboptimal habitats [22]. In such cases, the temporal frequency with which a species is observed might be a poor indicator of the extent to which a species can actually maintain a viable population in that location.

Understanding the frequency of classification errors and the factors that affect those errors is critical for properly interpreting patterns based on temporal occupancy. Here, we use a simulation-based approach to examine community dynamics—based on death, birth, dispersal, and establishment—on complex, dual-habitat landscapes in which species' habitat associations are known. We varied average species' detectability and habitat heterogeneity of the simulated landscapes to assess how these variables affect rates of misclassification. We expect that core species are more likely to be misclassified as transients when either detectability or abundance

**Table 1. Ways that species can be correctly or incorrectly (boxes in red) classified as maintaining a viable population based on temporal occupancy.**

| | Maintains a viable population $R_0 \geq 1$, "core" | Does not maintain a viable population $R_0 < 1$, "transient" |
|---|---|---|
| Low temporal occupancy, inferred to be "transient" | **A**: Species that occur persistently at low density or that have traits making them difficult to detect | **B**: Species that only irregularly occur in the local habitat because they are poorly suited to that habitat |
| High temporal occupancy, inferred to be "core" | **C**: Core members of the community that maintain viable populations and are reliably observed almost every year | **D**: Species that occur regularly in the local habitat despite failing to maintain positive population growth rates due to repeated immigration from adjacent source habitat |
| Error rates | A / (A + C) | D / (B + D) |

$R_0$ refers to the net reproductive rate of a species in a location.

is low. In contrast, we expect that species that do not successfully breed in a habitat are more likely to be misclassified as core members when the local community is embedded within a more heterogeneous landscape, which increases the likelihood of mass effects from adjacent habitats.

## Methods

### Simulation model

Each simulation began by generating an initial landscape, species pool, and global species abundance distribution (GSAD). The 32 x 32 pixel landscape was made up of two distinct habitat types, A and B, with a parameter for the proportion of the landscape made up of habitat type A ($h_A$; Fig 1A). Each grid cell represented a local community with a fixed community carrying capacity of 100 total individuals of any species. The species pool contained 40 total species, with half that could only reproduce successfully in habitat A and half that could only reproduce successfully in habitat B. The GSAD was a vector of relative species abundances assigned from a lognormal distribution that defined the relative probability that an immigrant from outside the landscape would belong to each species. Initially, the landscape was filled to carrying capacity with individuals drawn randomly from the GSAD.

In each time step, meant to represent one year, the following four processes were modeled:

1. *Death.* The probability of mortality for each individual at a time step was 0.5 (Fig 1B). Death rates were independent of the habitat type in which the species occurred.

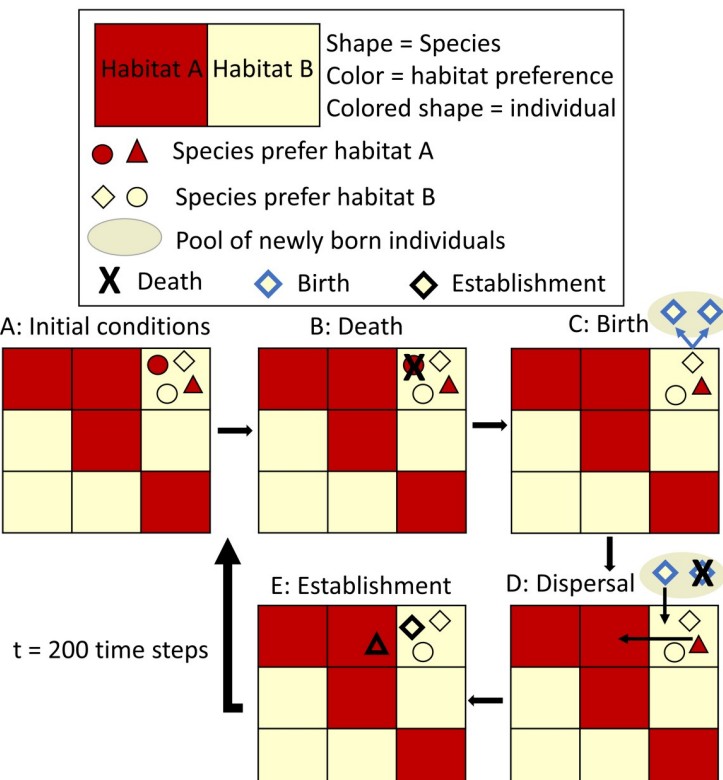

**Fig 1. Schematic documenting the events that occur in a single time step of the simulation, including death, birth, dispersal, and establishment.** See text for details.

2. *Birth*. All individuals occurring within their preferred habitat type produced two offspring per time step, while individuals occurring in a non-preferred habitat type did not reproduce. Offspring were termed "propagules" until they established in a community (see below; Fig 1C).

3. *Dispersal*. Newly generated propagules dispersed in random directions by a distance drawn from a half-Gaussian distribution with a mean of 1.24 grid cells (99% of movements result in dispersal distances ≤ 4 grid cells; Fig 1D). Established individuals (i.e. adults) only dispersed if they were in non-preferred habitats. We also explored dispersal kernels that were narrower (99% of movements within 2 grid cells) or broader (99% of movements within 8 grid cells) to confirm that results were qualitatively similar. Results for these simulations are presented in S1–S4 Figs, S1 and S2 Tables.

4. *Establishment*. Empty spaces in each community were colonized by either a migrant from outside the community (drawn probabilistically from the GSAD) with a constant immigration rate probability (0.001) or by an individual selected randomly from the pool of new or dispersing propagules. Once individuals became established, they only left their community via dispersal or death (Fig 1E). Propagules that did not establish were eliminated at the end of each time step.

We ran simulations for 200 time steps, which was long enough for species richness to achieve equilibrium in the landscape, and used the last 15 time steps to calculate temporal occupancy. Fifteen time steps represented an ecological dataset with a 15-year time series, a sampling period used in several previous studies which provides a reasonably high resolution estimate of temporal occupancy [7, 23]. Additionally, we calculated landscape-wide abundances for each species at the end of the simulation.

We ran 50 replicate simulations for values of $h_A \in \{0.5, 0.6, 0.7, 0.8, 0.9\}$ to generate landscapes that were more (high $h_A$) or less (low $h_A$) homogeneous. For each simulation, we also imposed a stochastic detection process in which we varied the probability of detecting an individual known to be present, $p$, from 0.1 to 1.0 in increments of 0.1. Detection probability was assumed to be both species- and habitat-independent. This resulted in a vector of "observed" species abundances in each grid cell at each time step.

## Simulation analysis

We examined the temporal dynamics of species within a single, centrally located pixel for each simulation run. Based on the habitat type of the focal pixel, all species either could (core) or could not (transient) reproduce within that pixel and hence maintain a viable population. We refer to this as their biological, or true, status. In addition, each species was classified as core or transient based on temporal occupancy over the last 15 years of the simulation run. Species observed in five years or fewer (≤ 33%) were classified as transient while species observed in more than ten years (> 66%) were classified as core. For these analyses, we ignored the minority of species with intermediate temporal occupancy which could not be unambiguously assigned to core or transient status. Thus, each of the species we considered fell into one of the four categories shown in Table 1. For each simulation run, we calculated the rate of misclassifying core species and the rate of misclassifying transient species (Table 1). Error rates were examined as a function of average detection probability and landscape similarity in the 7 x 7 pixel region surrounding the focal pixel, which was calculated as the proportion of the regional window that was the same habitat type as the focal pixel. We used this regional window size because it reflects the area over which most colonization events to the focal pixel would originate. Number of species and classification error rates were predicted by detection probability

and landscape similarity using ordinary least squares linear models. The relationship between species abundance and core species classification at detection = 0.5 was assessed using a generalized linear model with a logit link.

Code for running these simulations in R is archived at https://github.com/ssnell6/CT-sim.

## Results

Communities in homogeneous landscapes (e.g., Fig 2a) typically had a large number of true core species and only a few true transient species at any given point in time (Fig 2b). Turnover in the identity of the transient species from one time step to the next resulted in a mode of low temporal occupancy within an overall bimodal distribution of temporal occupancy (Fig 2c). Communities in heterogeneous landscapes (e.g., Fig 2d) had more true transient species appear in their non-preferred habitat type in any given time step due to the greater area of potential sources of colonization (Fig 2e). Many of these transient species were maintained by repeated dispersal from the alternate habitat type in the surrounding landscape such that they had moderate to high values of temporal occupancy (Fig 2f).

Due to the large number of simulation replicates run, all statistical relationships examined had p < 2e-16 (S1 and S2 Tables), so we focus here on reporting only the sign of effects and

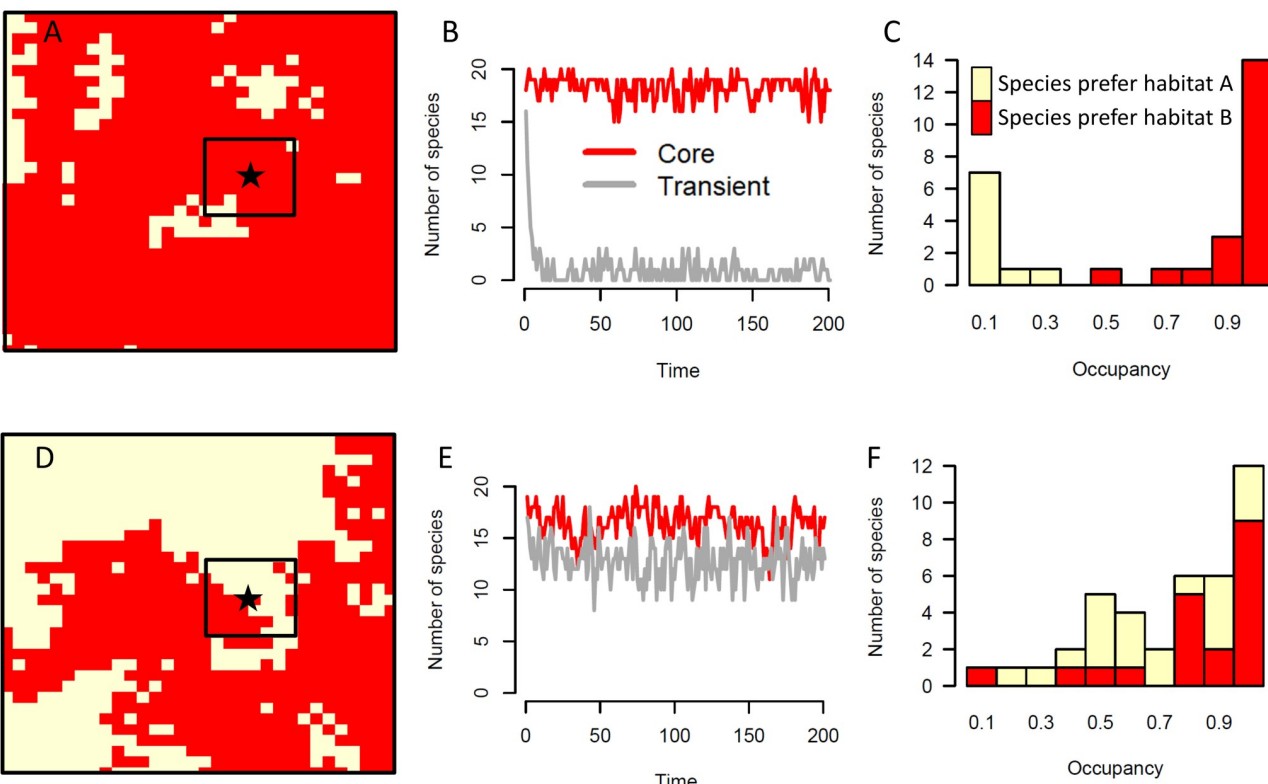

**Fig 2.** (A) Sample landscape of one simulation run in which the proportion of the full landscape that was habitat A (in red) was set to 0.9. Landscape similarity around the focal pixel is 0.92. (B) Number of core species (that can reproduce in the red habitat, red line) and transient species (that cannot reproduce in the red habitat, gray line), plotted over time for the focal pixel from the landscape in (A). (C) Temporal occupancy distribution of the species in the focal pixel from the landscape in (A). Colors of the bars indicate the number of species according to which habitat type they can reproduce in. (D) Sample landscape of one simulation run in which the proportion of the landscape that was habitat A (red) was set to 0.5. Landscape similarity around the focal pixel is 0.49. (E) Number of core species (red line) and transient species (gray line), plotted over time for the focal pixel from the landscape in (D). (F) Temporal occupancy distribution of the species in the focal pixel from the landscape in (D). Colors of the stacked bars indicate the number of species according to which habitat type they can reproduce in.

the variance explained. The number of true core species (those maintaining a locally viable population) observed in a pixel increased with detection probability, and even more so with landscape similarity (S5–S7 Figs). More variance in the number of true core species could be explained by landscape similarity ($R^2$ = 36%) than detection probability ($R^2$ = 2%). The number of true transient species (those not maintaining a viable population) observed increased with detection probability and decreased strongly with landscape similarity (S5–S7 Figs). More variance in the number of true transient species could be explained by landscape similarity ($R^2$ = 74%) than detection ($R^2$ = 5%).

Species that were true core members of the focal community were more likely to be incorrectly inferred as transient at low detection probabilities and low landscape similarities (Fig 3a). More variance in the proportion of misclassified core species could be explained by detection ($R^2$ = 46%) than landscape similarity ($R^2$ = 11%). Error rates were close to zero when landscape similarity was greater than 0.6 and detection probability was greater than 0.3 and increased most noticeably when detection probability was 0.1, the lowest detection rate examined.

Transient species that did not reproduce in the focal habitat but regularly occurred there were incorrectly inferred as core most often at high detection probabilities and low landscape similarities (Fig 3b). More variance in the proportion of misclassified transient species could be explained by landscape similarity ($R^2$ = 48%) than detection ($R^2$ = 13%). Error rates for classifying transient species were zero or near zero when landscape similarity was greater than 0.5. Transient species misclassification rates were greatest when landscape similarity was less than 0.4, where the majority of colonization events came from the opposite habitat type, such that poorly adapted species appeared in the focal habitat repeatedly over time. This was exacerbated at high detection probability, which ensured these true transient occurrences were observed and therefore misclassified. Additionally, species with low landscape-wide abundance were more likely to be misclassified as transient when they were truly core members of their

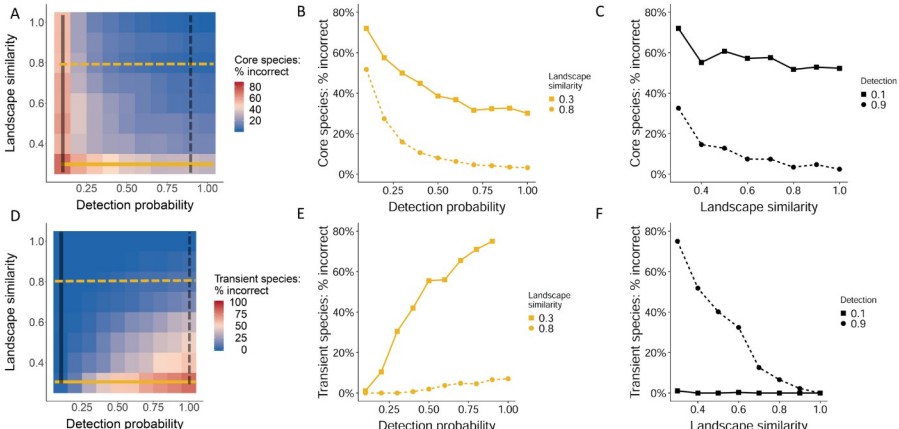

**Fig 3.** Percent of biologically core (A) species that were incorrectly inferred to be transient and biologically transient (D) species that were incorrectly inferred to be core for each combination of detection probability and landscape similarity. The x-axis is the average species detection probability for the simulation run, while the y-axis is the proportion of a 7 x 7 landscape surrounding the focal pixel that is of the same habitat type. Line graphs (B, E) show the percent of incorrect classifications of core species (B) or transient species (E) for each detection probability at low (0.3, solid line) or high (0.8, dashed line) landscape similarity. Line graphs (C, F) show the percent of incorrect classifications of core species (C) or transient species (F) with increasing landscape similarity at low (0.1, solid line) or high (0.9, dashed line) detection probability.

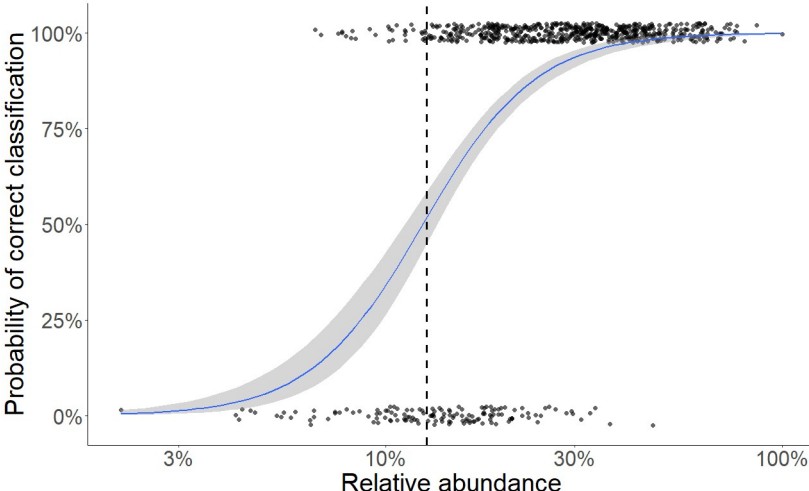

**Fig 4. Probability of correct classification of biologically core species based on temporal occupancy as a function of the log of landscape wide abundance (relative to the abundance of the most abundant species, 100%).** Dashed line indicates the location of the inflection point.

community, while the odds of misclassifying a core species were less than 13% for species whose abundance was at least 12% of the most abundant species (Fig 4).

Results were similar using both narrower and broader dispersal kernels (S1–S4 Figs, S1 and S2 Tables), with narrow kernels having slightly more variance in classification rate than broader kernels.

## Discussion

Several studies have used temporal occupancy to infer the persistence of populations over time and the degree to which a species can be considered a core member of a community in a particular location [3, 7–9, 23]. Our simulations showed that in many realistic scenarios, this is a valid approach, but also confirmed that temporal occupancy is subject to misclassification errors where core species are inferred to be transient and transient species are inferred to be core. As expected, low detection probabilities resulted in more frequent misclassification of core species as transient. Rare species were also more likely misclassified as transient. Low landscape similarity, when combined with high detection probabilities, resulted in transient species more frequently being misclassified as core.

Imperfect individual detection influenced the rate at which core species were misclassified as transients through failing to detect species when they were actually present. These species were more likely to be inferred as transient at lower detection probabilities. However, error rates for core species misidentified as transients were quite low as long as detection probabilities were greater than approximately 0.3. This threshold of 0.3 is at the low end of detection probabilities observed for most bird species, with most species exhibiting substantially higher rates of detection [24–26]. Specifically, Boulinier *et al.* (1998) found that across a range of habitats in North America, average detection probabilities for species richness estimates using the Breeding Bird Survey ranged from 0.65 to 0.85. Johnston et *al.* (2014) found that the least detectable family of birds was *Paridae*, which had a median detectability of 0.27, and the majority of other families had detection probabilities greater than 0.3. Overall, these findings suggest that the misclassification of core species is unlikely to be common except at unusually low detection probabilities that may be relevant for only a small minority of species.

Misclassification of core species as transients was also more common for species occurring at low abundance across the landscape, and in particular, for species with abundances less than 12% of the most abundant species. The probability of detecting a species with $n$ individuals given an individual detection probability, $p$, will be $1 - (1 - p)^n$, and thus this link between abundance, species detection, and potential misclassification of transient status is quite expected. Also, species that occur at low density may have large home ranges relative to the scale of the survey (e.g. woodpeckers, raptors), and may frequently be missed on surveys even if they are on territory and have high detectability when present. This is one reason that relatively small spatial scales have been shown to have fewer perceived core species and more perceived transient species compared to larger scales [3, 27].

The opposite classification error, misclassifying transient species as core species, was associated with high habitat heterogeneity. Species occurred in habitats to which they were poorly adapted because of dispersal from nearby source populations. The greater the surrounding area containing source populations, the greater the chance of repeated dispersal into nearby sink habitats causing the species to be regularly detected through time [20]. These errors became prevalent when 60% or more of the surrounding landscape was different from the focal habitat. Our simulation model assumed that dispersal of new propagules was random with respect to habitat type, but if dispersal was biased toward the preferred habitat type (which seems likely for organisms with active dispersal; i.e., Johnston et al. 2014), it would reduce the frequency of transient occurrences and therefore reduce observed error rates. The rate at which transient species were misclassified as core species also decreased with decreasing detection probability because at low detections, errors caused by repeated dispersal from adjacent source habitats were canceled out by detection errors. Overall, these results suggest that misclassification of transient species is unlikely to be common except in highly fragmented landscape configurations with unbiased dispersal.

Geographic patterns in the relative prevalence of core and transient species can influence our understanding of ecological communities when failing to recognize this distinction [7, 23], especially if the probability of misclassification varies geographically. One likely source for this is detection probability, which is thought to vary along environmental gradients. In particular, it has been suggested that average detectability decreases along continental to global productivity gradients because species are more difficult to detect in more densely vegetated environments [17]. However, despite such a potential bias, past work has shown that there is typically a positive relationship between either temporal occupancy or species richness and remotely sensed proxies for productivity, meaning the observed patterns were opposite what would be predicted purely from a detectability effect [7, 17, 23]. As such, these patterns of occupancy and richness were observed despite, and not because of, geographic variation in detectability. Other studies have suggested that birds sing more frequently in densely forested habitats so aural-based sampling should not observe this effect in forests, but in open habitats [24]. In these cases, variation in detection probability alone has the potential to generate apparent patterns in richness or abundance, with misclassification rates of species varying across the gradient.

While we parameterized our simulation model to loosely reflect the biology of songbirds (e.g. reproductive rate, dispersal distance), the inferences that can be made from this simulation model are more broadly generalizable. We chose to focus on birds because they are highly mobile, can disperse widely, and have been studied empirically in this core-transient context [3, 7, 23]. These first two attributes make temporal occupancy particularly useful for identifying core and transient birds in communities, but also potentially more prone to errors due to source-sink dynamics.

Detectability is dependent on both species attributes and the environment. Some species are inherently more detectable due to variation in species color, size, and behavior. A large, colorful bird perched conspicuously or that sings loudly and frequently is detected more often than a little brown bird in the undergrowth, given they occur at equal densities. Our study is most relevant for considering how detection probability covaries along an environmental gradient, where detection probability likely varies on average across all species, than for considering how detection probability varies among species. Nevertheless, species known to have low detection probabilities will presumably require more targeted monitoring efforts, and temporal occupancy should be used with caution to infer population persistence and habitat suitability for such species.

The aim of our simulation model was to capture how frequently species are misclassified within the core-transient temporal occupancy framework. Therefore, we focused on landscape similarity and detectability, but other parameters could also play a role in determining the effectiveness of temporal occupancy. In our study, birth rates and death rates were constant, so increasing the birth rates or decreasing death rates of species occurring in their preferred habitats could allow specialists to reach equilibrium in a habitat more quickly, decreasing the number of transient species in the community. Additionally, varying immigration across species could enable one species to immigrate more effectively into new habitats than other species, but in our study, immigration and dispersal were analogous because both allowed species to colonize new habitats. We addressed alternative dispersal distances into new cells by varying the dispersal kernels in supplementary analyses, which demonstrated that only at very low dispersal do detection and landscape similarity affect core and transient classification.

In general, we found that temporal occupancy can reliably be used to infer habitat associations, as well as the likelihood of a species maintaining a viable population in the location where it was observed, under a broad range of conditions. Depending on the nature of the raw data available, occupancy modeling approaches (sensu MacKenzie et al. 2002) may have the potential to refine assignments of core and transient species status by directly accounting for detectability, and deserve further research in this context. The use of raw temporal occupancy may be most problematic in study systems made up of highly isolated habitat fragments where species commonly disperse from the surrounding landscape matrix, or in habitats or for species with uniformly low detection probabilities. Ecologists should explicitly consider whether detection probabilities vary across the environmental gradients in their study systems before using temporal occupancy. Considering the relationship of landscape similarity and detection in specific study systems will provide a guide for when and how to include temporal occupancy in ecological analyses.

## Supporting information

**S1 Fig.** Percent of biologically core (A) species that were incorrectly inferred to be transient and biologically transient (D) species that were incorrectly inferred to be core for each combination of detection probability and landscape similarity at a narrower dispersal kernel (99% of movements result in dispersal distances ≤ 2 grid cells). The x-axis is the average species detection probability for the simulation run, while the y-axis is the proportion of a 7 x 7 landscape surrounding the focal pixel that is of the same habitat type. Line graphs (B, E) show the percent of incorrect classifications of core species (B) or transient species (E) for each detection probability at low (0.3, solid line) or high (0.8, dashed line) landscape similarity. Line graphs (C, F) show the percent of incorrect classifications of core species (C) or transient species (F) with

increasing landscape similarity at low (0.1, solid line) or high (0.9, dashed line) detection probability.
(DOCX)

**S2 Fig. Correct or incorrect classification of biologically core species based on temporal occupancy as a function of the log of landscape wide abundance (relative to the abundance of the most abundant species) at a narrower dispersal kernel (99% of movements result in dispersal distances ≤ 2 grid cells).**
(DOCX)

**S3 Fig.** Percent of biologically core (A) species that were incorrectly inferred to be transient and biologically transient (D) species that were incorrectly inferred to be core for each combination of detection probability and landscape similarity at a broader dispersal kernel (99% of movements result in dispersal distances ≤ 8 grid cells). The x-axis is the average species detection probability for the simulation run, while the y-axis is the proportion of a 7 x 7 landscape surrounding the focal pixel that is of the same habitat type. Line graphs (B, E) show the percent of incorrect classifications of core species (B) or transient species (E) for each detection probability at low (0.3, solid line) or high (0.8, dashed line) landscape similarity. Line graphs (C, F) show the percent of incorrect classifications of core species (C) or transient species (F) with increasing landscape similarity at low (0.1, solid line) or high (0.9, dashed line) detection probability.
(DOCX)

**S4 Fig. Correct or incorrect classification of biologically core species based on temporal occupancy as a function of the log of landscape wide abundance (relative to the abundance of the most abundant species) at a broader dispersal kernel (99% of movements result in dispersal distances ≤ 8 grid cells).**
(DOCX)

**S5 Fig.** Mean number of biologically core (A) and biologically transient (D) species observed for each combination of detection probability and landscape similarity at the main analysis dispersal kernel (99% of movements result in dispersal distances ≤ 4 grid cells). Line graphs (B, E) show the mean count of core species (B) or transient species (E) for each detection probability at low (0.3, solid line) or high (0.8, dashed line) landscape similarity. Line graphs (C, F) show the mean count of core species (C) or transient species (F) with increasing landscape similarity at low (0.1, solid line) or high (0.9, dashed line) detection probability.
(DOCX)

**S6 Fig.** Mean number of biologically core (A) and biologically transient (D) species observed for each combination of detection probability and landscape similarity at a narrower dispersal kernel (99% of movements result in dispersal distances ≤ 2 grid cells). Line graphs (B, E) show the mean count of core species (B) or transient species (E) for each detection probability at low (0.3, solid line) or high (0.8, dashed line) landscape similarity. Line graphs (C, F) show the mean count of core species (C) or transient species (F) with increasing landscape similarity at low (0.1, solid line) or high (0.9, dashed line) detection probability.
(DOCX)

**S7 Fig.** Mean number of biologically core (A) and biologically transient (D) species observed for each combination of detection probability and landscape similarity at a broader dispersal kernel (99% of movements result in dispersal distances ≤ 8 grid cells). Line graphs (B, E) show the mean count of core species (B) or transient species (E) for each detection probability at low (0.3, solid line) or high (0.8, dashed line) landscape similarity. Line graphs (C, F) show the

mean count of core species (C) or transient species (F) with increasing landscape similarity at low (0.1, solid line) or high (0.9, dashed line) detection probability.
(DOCX)

**S1 Table. Parameter estimates from linear models of the number of core and transient species as a function of detection and landscape similarity for all dispersal kernels.**
(DOCX)

**S2 Table. Parameter estimates from linear models of the percent incorrect core and transient species as a function of detection and landscape similarity for all dispersal kernels.**
(DOCX)

# Acknowledgments

Thank you to K. Eklund and J. Roach for assistance and feedback on the analytic workflow using a computing cluster.

# Author Contributions

**Conceptualization:** Jessica R. Coyle, Ethan P. White, Allen H. Hurlbert.

**Data curation:** Jessica R. Coyle, Ethan P. White, Allen H. Hurlbert.

**Formal analysis:** Sara Snell Taylor.

**Funding acquisition:** Ethan P. White, Allen H. Hurlbert.

**Investigation:** Sara Snell Taylor, Jessica R. Coyle, Ethan P. White, Allen H. Hurlbert.

**Methodology:** Sara Snell Taylor, Jessica R. Coyle.

**Project administration:** Allen H. Hurlbert.

**Resources:** Allen H. Hurlbert.

**Supervision:** Allen H. Hurlbert.

**Validation:** Sara Snell Taylor.

**Visualization:** Sara Snell Taylor.

**Writing – original draft:** Sara Snell Taylor.

**Writing – review & editing:** Sara Snell Taylor, Jessica R. Coyle, Ethan P. White, Allen H. Hurlbert.

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
