## [Decision Letter · Decision Letter 0]

25 Aug 2020

PONE-D-20-24435

A simulation study of the use of temporal occupancy for identifying core and transient species

PLOS ONE

Dear Dr. Snell Taylor,

Thank you for submitting your manuscript to PLOS ONE. After careful consideration, we feel that it has merit but does not fully meet PLOS ONE’s publication criteria as it currently stands. Therefore, we invite you to submit a revised version of the manuscript that addresses the points raised during the review process.

Both reviewers found the study interesting and generally well written. However both reviewers also felt some additional justification of the modelling framework is needed, and found the distinction between transient and sink species somewhat unclear.   

We look forward to receiving your revised manuscript.

Kind regards,

Patrick R Stephens, Ph.D.

Academic Editor

PLOS ONE

Reviewers' comments:

Reviewer's Responses to Questions

**Comments to the Author**

1. Is the manuscript technically sound, and do the data support the conclusions?

Reviewer #1: Yes

Reviewer #2: Yes

2. Has the statistical analysis been performed appropriately and rigorously? 

Reviewer #1: Yes

Reviewer #2: Yes

3. Have the authors made all data underlying the findings in their manuscript fully available?

Reviewer #1: Yes

Reviewer #2: Yes

4. Is the manuscript presented in an intelligible fashion and written in standard English?

Reviewer #1: Yes

Reviewer #2: Yes

5. Review Comments to the Author

Reviewer #1: This paper uses a simulation model to examine the roles that heterogeneity in both landscapes and detection probability influence categorization of species as either 'core' or 'transient' members of a community. The results of the model are in line with the authors expectations that core species will be misidentified as transient mostly due to lower detection probabilities, and that transient species will be misidentified as core mostly due to increased landscape heterogeneity. These results, though not too surprising, reinforce the notion that mis-classification of species is possible, and quantifies under which scenarios different mis-classifications are expected to occur.

The model itself is well-described in the Methods section, and the subheadings make it easy to follow this description. The code is available in the supplemental material. The Methods and Results are written very clearly and succinctly, and would be possible to reproduce in any programming language. All in all, I thought this paper flowed very naturally.

I have just a few questions:

1) Are 'transient' and 'sink' species the same thing (line 54)? How does this model compare or add to the source/sink dynamic models?

2) Is landscape heterogeneity the major driver of transient species occurrences in communities? I'm not too familiar with the literature on core/transient species in communities, but it seems like factors other than habitat preference might also drive the occurrence of a transient species in a community. For example, naturally low population sizes, meta-population dynamics, etc. This is beyond the scope of this model, but it would be good to mention other factors that influence whether a species is transient.

3) Most of the model parameters make sense and are explained clearly. But why did you choose a 7x7 pixel region surrounding the focal pixel to quantify heterogeneity? Did you make sure that this sized region was truly reflective of landscape heterogeneity on the whole grid? Given enough simulations, this might not be an issue, but maybe a little explanation about why you chose this size would be helpful.

Reviewer #2: This is a simulation study that aims to clarify potential biases associated with applying temporal occupancy approaches to classify core and transient species. While the results are somewhat intuitive, as misclassification depends on both detection rates and the spatial proximity between habitats, putting these recommendations “out there” will be useful for many ecologists. I have a few suggestions, but I feel they are relatively minor.

Sincerely,

Jonathan Belmaker

Introduction:

The ability to detect core or transient species will depend on how temporal occupancy is used to separate these groups. A few more words here would be useful for the naïve reader to understand how core and transient species are separated in other studies, and what approach will be used here.

Methods:

Very specific birth, death, immigration rate and carrying capacity parameters were used. I do not believe this will change the results, but some sensitivity analyses to the modification of these parameters would increase the robustness of the findings (this was done for dispersal rate only).

For the simulations varying detection probability, the distribution of detection probabilities across species and habitats is not clear. Please clarify.

I had to get to the discussion to understand that detection probability did not vary among species. I would like to better understand the rational for this, as it seems less relevant to natural communities than models that vary detection among species. Furthermore, I would expect that when detection rates vary among species the misclassification of core and transient species may increase dramatically. I would strongly suggest to explore adding such heterogeneity in detection rate to the simulations.

Results:

It does not make sense to use P values in simulation models (as one can always achieve a significant result by increasing sample size). I would delete the P values throughout the text.

Discussion:

Occupancy modeling should improve imperfect detection. It is worth discussing how these methods may improve the separation between core and transient species given the results presented here.

The impact of rarity is strong and nicely seen in figure 5, but I feel does not receive sufficient attention in the discussion.

Figures and tables:

I am not sure Figure 3 is very informative, as it is not the number of core and transient species that is interesting in this context but the misclassification rates. I would remove.

6. PLOS authors have the option to publish the peer review history of their article (what does this mean?). If published, this will include your full peer review and any attached files.

Reviewer #1: **Yes: **Andrew Laughlin

Reviewer #2: **Yes: **Jonathan Belmaker

---

## [Author Response · Author response to Decision Letter 0]

25 Sep 2020

For formatted responses, please see attached document.

I have just a few questions:

1) Are 'transient' and 'sink' species the same thing (line 54)? How does this model compare or add to the source/sink dynamic models?

Although we link the processes we describe to source-sink dynamics throughout the text, we agree that it was unclear to refer to the term "sink species", as “sink” is better used to describe the location than the species occurring in that location. To clarify, we have deleted the “or sink” phrase in this sentence.

2) Is landscape heterogeneity the major driver of transient species occurrences in communities? I'm not too familiar with the literature on core/transient species in communities, but it seems like factors other than habitat preference might also drive the occurrence of a transient species in a community. For example, naturally low population sizes, meta-population dynamics, etc. This is beyond the scope of this model, but it would be good to mention other factors that influence whether a species is transient.

Indeed, low population density and metapopulation dynamics do contribute to the likelihood of a species being classified as transient. We mention low population density on line 67 of the introduction and have expanded our discussion of this effect on lines 230-240 of the discussion. Metapopulation dynamics occur due to discrete populations connected by varying degrees by dispersal across a heterogeneous landscape. This is what we are explicitly modeling and what our simulated landscape with varying degrees of heterogeneity seeks to represent.

3) Most of the model parameters make sense and are explained clearly. But why did you choose a 7x7 pixel region surrounding the focal pixel to quantify heterogeneity? Did you make sure that this sized region was truly reflective of landscape heterogeneity on the whole grid? Given enough simulations, this might not be an issue, but maybe a little explanation about why you chose this size would be helpful.

Thank you for this comment. Each pixel on the landscape contains an entire biological community of organisms and we wanted to represent the heterogeneity of the region over which the vast majority of dispersing propagules are colonizing from. For the main dispersal kernel we examined, 99% of movements occurred within 4 grid cells. Had we chosen a larger window, landscape features outside of the likely range of colonization would influence our estimate of heterogeneity. We have added a sentence justifying this decision in lines 154-155. 

Reviewer #2: 

Introduction:

The ability to detect core or transient species will depend on how temporal occupancy is used to separate these groups. A few more words here would be useful for the naïve reader to understand how core and transient species are separated in other studies, and what approach will be used here.

We have added text and citations to clarify how core and transient species have previously been separated in the introduction (ll. 59-61).

Methods:

Very specific birth, death, immigration rate and carrying capacity parameters were used. I do not believe this will change the results, but some sensitivity analyses to the modification of these parameters would increase the robustness of the findings (this was done for dispersal rate only).

We appreciate the reviewer’s suggestions for sensitivity analyses, and we carried out this analysis for dispersal rate, which, of all the model parameters, is most directly connected to the process leading to the presence of transient species in habitats to which they are unsuited. The other parameters mentioned by the reviewer (birth rate, death rate, immigration rate, and carrying capacity) are largely independent of the dispersal and detection processes in our model that would affect the misclassification of species as core or transient. As we mention in the discussion (lines 287-298), interspecific variability in these parameters and others might be interesting to consider for other reasons, but interspecific variation is beyond the scope of our present aims. 

For the simulations varying detection probability, the distribution of detection probabilities across species and habitats is not clear. Please clarify.

I had to get to the discussion to understand that detection probability did not vary among species. 

On line 137-139 of the methods section, we state that “Detection probability was assumed to be both species- and habitat-independent.”

I would like to better understand the rational for this, as it seems less relevant to natural communities than models that vary detection among species. Furthermore, I would expect that when detection rates vary among species the misclassification of core and transient species may increase dramatically. I would strongly suggest to explore adding such heterogeneity in detection rate to the simulations.

We appreciate the reviewer’s comment regarding detection probability, and they are absolutely correct that detectability varies among species empirically based on traits such as color, size, and behavior. Introducing interspecific variation in detection probabilities would undoubtedly help explain which species are being misclassified within the simulation, but this is not our aim and we take it as a given. We disagree with the reviewer’s intuition regarding the effect of heterogeneity in detectability across species within a simulation run. The fewest misclassifications of species as transient will occur when detectability is high for all species, the most misclassifications will occur when detectability is low for all species, and when there is a mix of high and low detectability species, the misclassification rate for the community will be intermediate.

Instead, we are interested in the consequences of imperfect detectability across an entire community, given that detectability may vary on average from community to community as a function of environmental characteristics. We have added text in the introduction on lines 71-73 to emphasize this motivation. Given these aims, and the fact that a “data point” in our analyses is a grid cell (characterized by mean detectability and landscape heterogeneity) and not a species, we find it most appropriate to vary detectability as we have which provides a sense of the best and worst case scenarios. 

Results:

It does not make sense to use P values in simulation models (as one can always achieve a significant result by increasing sample size). I would delete the P values throughout the text. 

Thank you for pointing this out. We now include a statement acknowledging this fact lines 171-173), and we no longer refer to the P values in the results section of the manuscript.

Discussion:

Occupancy modeling should improve imperfect detection. It is worth discussing how these methods may improve the separation between core and transient species given the results presented here.

We have added the following sentence to the Discussion (lines 302-305):

"Depending on the nature of the raw data available, occupancy modeling approaches (sensu MacKenzie et al. 2006) may have the potential to refine assignments of core and transient species status by directly accounting for detectability, and deserve further research in this context."

That said, there are many nuances regarding the types of data that occupancy modeling requires and the specific questions that it is able to answer, and we feel this extended discussion is tangential to the aims of our manuscript. Specifically, we would argue that “occupancy” as used in the occupancy estimation modeling literature is typically a description of spatial distribution, defined in MacKenzie et al. (2006) as the “proportion of an area occupied by a species or fraction of landscape units where the species is present.” Proper occupancy modeling requires repeat visits within a year in order to estimate detectability and in datasets where sites are visited only once per year, it is more difficult to separate detectability and occupancy. Temporal occupancy is instead a proxy for population persistence over time that can be calculated for a much broader array of dataset types, and the point of this manuscript is that it performs reasonably well. 

The impact of rarity is strong and nicely seen in figure 5, but I feel does not receive sufficient attention in the discussion.

 Thank you for pointing this out. We have added a paragraph discussing this result on lines 230-240:

"Misclassification of core species as transients was also more common for species occurring at low abundance across the landscape, and in particular, for species with abundances less than 12% of the most abundant species. The probability of detecting a species with n individuals given an individual detection probability, p, will be 1 – (1 – p)n, and thus this link between abundance, species detection, and potential misclassification of transient status is quite expected. Also, species that occur at low density may have large home ranges relative to the scale of the survey (e.g. woodpeckers, raptors), and may frequently be missed on surveys even if they are on territory and have high detectability when present. This is one reason that relatively small spatial scales have been shown to have fewer perceived core species and more perceived transient species compared to larger scales (Snell Taylor et al. 2018; Jenkins et al. 2018)."

Figures and tables:

I am not sure Figure 3 is very informative, as it is not the number of core and transient species that is interesting in this context but the misclassification rates. I would remove.

We have moved Figure 3 to the supplemental information (Figure S5). We feel it remains useful for contextualizing the data on misclassification rates in Figure 4 given that there is variation in the total number of transient and core species that could potentially be misclassified across parameter space.

---

## [Decision Letter · Decision Letter 1]

12 Oct 2020

A simulation study of the use of temporal occupancy for identifying core and transient species

PONE-D-20-24435R1

Dear Dr. Snell Taylor,

We’re pleased to inform you that your manuscript has been judged scientifically suitable for publication and will be formally accepted for publication once it meets all outstanding technical requirements.

Kind regards,

Patrick R Stephens, Ph.D.

Academic Editor

PLOS ONE

Additional Editor Comments (optional):

Reviewers' comments:

Reviewer's Responses to Questions

**Comments to the Author**

1. If the authors have adequately addressed your comments raised in a previous round of review and you feel that this manuscript is now acceptable for publication, you may indicate that here to bypass the “Comments to the Author” section, enter your conflict of interest statement in the “Confidential to Editor” section, and submit your "Accept" recommendation.

Reviewer #1: All comments have been addressed

Reviewer #2: All comments have been addressed

2. Is the manuscript technically sound, and do the data support the conclusions?

Reviewer #1: Yes

Reviewer #2: Yes

3. Has the statistical analysis been performed appropriately and rigorously? 

Reviewer #1: Yes

Reviewer #2: Yes

4. Have the authors made all data underlying the findings in their manuscript fully available?

Reviewer #1: Yes

Reviewer #2: Yes

5. Is the manuscript presented in an intelligible fashion and written in standard English?

Reviewer #1: Yes

Reviewer #2: Yes

6. Review Comments to the Author

Reviewer #1: (No Response)

Reviewer #2: This is the second time I am reviewing this manuscript. In general, my previous comments were addressed (for a few small issues I do not completely agree, but this is a matter of opinion and I am willing to accept the authors’ response). I do not have any new suggestions I will be happy to see this accepted for publication.

Sincerely,

Jonathan Belmaker

7. PLOS authors have the option to publish the peer review history of their article (what does this mean?). If published, this will include your full peer review and any attached files.

Reviewer #1: **Yes: **Andrew Laughlin

Reviewer #2: **Yes: **Jonathan Belmaker

---

## [Editor Report · Acceptance letter]

14 Oct 2020

PONE-D-20-24435R1 

A simulation study of the use of temporal occupancy for identifying core and transient species 

Dear Dr. Snell Taylor:

I'm pleased to inform you that your manuscript has been deemed suitable for publication in PLOS ONE. Congratulations! Your manuscript is now with our production department. 

Kind regards, 

on behalf of

Dr. Patrick R Stephens 

Academic Editor

PLOS ONE